# Pyrolysis and Gasification of a Real Refuse-Derived Fuel (RDF): The Potential Use of the Products under a Circular Economy Vision

**DOI:** 10.3390/molecules27238114

**Published:** 2022-11-22

**Authors:** Michela Alfè, Valentina Gargiulo, Michele Porto, Renata Migliaccio, Adolfo Le Pera, Miriam Sellaro, Crescenzo Pellegrino, Abraham A. Abe, Massimo Urciuolo, Paolino Caputo, Pietro Calandra, Valeria Loise, Cesare Oliviero Rossi, Giovanna Ruoppolo

**Affiliations:** 1CNR-STEMS, National Research Council, Institute of Sciences and Technologies for Sustainable Energy and Mobility, P.le V. Tecchio 80, 80125 Napoli, Italy; 2UdR INSTM–STEMS, P.le V. Tecchio 80, 80125 Napoli, Italy; 3Department of Chemistry and Chemical Technologies, University of Calabria, Via P. Bucci, Cubo 14/D, 87036 Rende, Italy; 4Calabra Maceri e Servizi s.p.a., Via M. Polo 54, 87036 Rende, Italy; 5CNR-ISMN, National Research Council, Institute of Nanostructured Materials, Via Salaria km 29.300, 00015 Montelibretti, Italy; 6UdR INSTM della Calabria, Via P. Bucci, Cubo 14/D, 87036 Rende, Italy

**Keywords:** refuse-derived fuel, pyrolysis, gasification, product yields, pyrolysis products, waxes, char, syngas

## Abstract

Refuse-Derived Fuels (RDFs) are segregated forms of wastes obtained by a combined mechanical–biological processing of municipal solid wastes (MSWs). The narrower characteristics, e.g., high calorific value (18–24 MJ/kg), low moisture content (3–6%) and high volatile (77–84%) and carbon (47–56%) contents, make RDFs more suitable than MSWs for thermochemical valorization purposes. As a matter of fact, EU regulations encourage the use of RDF as a source of energy in the frameworks of sustainability and the circular economy. Pyrolysis and gasification are promising thermochemical processes for RDF treatment, since, compared to incineration, they ensure an increase in energy recovery efficiency, a reduction of pollutant emissions and the production of value-added products as chemical platforms or fuels. Despite the growing interest towards RDFs as feedstock, the literature on the thermochemical treatment of RDFs under pyrolysis and gasification conditions still appears to be limited. In this work, results on pyrolysis and gasification tests on a real RDF are reported and coupled with a detailed characterization of the gaseous, condensable and solid products. Pyrolysis tests have been performed in a tubular reactor up to three different final temperatures (550, 650 and 750 °C) while an air gasification test at 850 °C has been performed in a fluidized bed reactor using sand as the bed material. The results of the two thermochemical processes are analyzed in terms of yield, characteristics and quality of the products to highlight how the two thermochemical conversion processes can be used to accomplish waste-to-materials and waste-to-energy targets. The RDF gasification process leads to the production of a syngas with a H_2_/CO ratio of 0.51 and a tar concentration of 3.15 g/m^3^.

## 1. Introduction

An ever-increasing pressure on resources and environmental protection, especially in CO_2_ reduction [1], leads to “a systemic change in the use and recovery of resources in the economy” through a clear transition to a regenerative circular economy [2,3] by creating a closed-loop system that minimizes the use of resource inputs and the creation of wastes, pollution and overall carbon emissions [3]. From this perspective, the reuse of opportunely treated municipal solid wastes (MSWs) represent a good starting point to fulfil the circular economy goals [4]. To promote waste treatment options in line with the EU waste hierarchy based on prevention, preparing for reuse, recycling, recovery and disposal, the European Commission has adopted a Circular Economy Package [5] that includes a ban on separately collected waste landfilling, a common EU target of 65% of recycled municipal waste and the reduction to maximum 10% of landfilled municipal waste by 2030.

The most attractive characteristic of the MSW is surely the possibility to extract the great amount of chemical energy stored in it. The calorific value which can be generated from municipal waste is around 8–12 MJ/kg [6]. Up to date incineration is the most widespread approach for the energy recovery from MSWs [7]. However, a number of post-treatments are necessary to eliminate the hazardous and toxic content from the ashes resulting from MSW incineration [8]. In addition, flue gas emissions from the incineration of MSWs contain several pollutants as particulates, carbon oxides, nitrogen oxides, sulfur oxides, hydrochloric acid, heavy metals, dioxins (PCDD) and furans (PCDF) that require expensive cleaning systems. It has been reported that each ton of incinerated MSWs can produce 15–40 kg of hazardous wastes [9]. The high variability and heterogeneity of MSW composition (e.g., seasonal, regional variability) [10] heavily influences the emissions of pollutants generated during incineration as well as the thermochemical process performances, leading to a nonuniform thermal behavior and poor process outcome reproducibility [11]. To overcome these difficulties and to respect the concepts of sustainability and the circular economy, based on material recycling and resource recovery, different MSW pretreatment approaches have been proposed. The aim of such pretreatments is to recover as many recyclable materials as possible, reduce the size of the feedstock and separate the combustible substances from noncombustible fraction and high-moisture materials (i.e., production of refuse-derived fuels, RDFs) [12].

MSWs consist of three major fractions: a combustible fraction, a noncombustible fraction and moisture or volatile material. The combustible fraction is separated from the other two by specific sorting facilities [12]. Typically, MSWs are processed to remove the recyclable fraction (e.g., metals), the inert fractions (e.g., glass) and separate the fine wet organic fraction (e.g., food and garden waste) containing high-moisture and high-ash material. There are two basic sorting methodologies which have been developed to produce MSWs-derived fuel [13,14]: the Mechanical Biological Treatment (MBT) and the Biological Drying Process. Since the RDF production concept ensures a certain degree of size reduction and the removal of organic and inert material, RDF is characterized, on average, by a higher heating value, lower ash content and a lower bulk density compared to untreated waste. As consequence, the composition [15] and characteristics [16] of RDF processed from MSWs showed narrower characteristics more suitable for thermochemical valorization purposes [17] such as high calorific value (18–24 MJ/kg), low moisture content (3–6%) and high volatile (77–84%) and carbon (47–56%) content. It has to be mentioned that the use of RDF as a source of energy is encouraged as it is an integral part of waste management and it is regulated by EU regulations (EU Parliament Directive 2008/98/EC on waste) [17].

Pyrolysis and gasification are emerging as promising thermochemical processes for MSW and RDF treatments. Compared to incineration, MSW pyrolysis and gasification are very attractive technologies as they increase energy recovery efficiency, reduce the dimension of the section of post-treatment for pollutants emissions control, prevent PCDD/PCDF formation (due to reducing conditions) and possibly generate value-added products as chemical platforms or fuels [18]. Pyrolysis is a thermal degradation process to convert a carbon-rich matrix into solid, liquid and gas products in the absence of oxygen and at elevated temperature (400–700 °C) [19]. Characteristics and yield of the pyrolysis products strongly depend on the choice of operating parameters such as process temperature, feedstock resident time, heating rate and volatiles residence time. Pressure is not expressly listed among the pyrolysis operating variables since it indirectly affects the volatiles residence time [20,21]. Depending on the operating conditions, the pyrolysis process can be classified as slow, conventional, fast or flash. In slow pyrolysis the heating rate is kept slow (approximately 0.1–1 °C/s) to obtain higher char yields at the expense of the other products. In conventional pyrolysis, moderate heating rates (up to 200–300 °C/min) are applied, leading to a more homogeneous distribution among the product yields. In fast pyrolysis, faster heating rates (between 10 and 200 °C/s) are applied to obtain high yields of liquid or gas products at the expense of a solid one. In flash pyrolysis, the heating rates are very high (>1000 °C/s) and the reaction times are of few to several seconds, allowing for the production of a high amount of liquid products [20,21].

Regarding the application of pyrolysis for waste treatment, many studies on the pyrolysis of MSWs and segregated MSWs are available [22,23,24,25], but it is necessary to underline that the pyrolysis process is more feasible for the treatment of homogeneous waste, which shows low variations in composition such as lignocellulosic biomasses. Indeed, the products obtained from the pyrolysis of MSW cannot be generalized because of the high variable thermal and chemical MSW compositions. Pyrolysis tests on MSWs and segregated MSWs have been mainly performed in fixed bed and tubular reactors, auger reactors, batch and semi batch reactors, fluidized-bed reactors and rotary kilns from lab scale up to pilot scale, mainly under slow and fast pyrolysis conditions. Very recently, microwave-assisted pyrolysis processes have also been proposed for MSW pyrolysis. In addition, in many cases catalytic pyrolysis processes are performed to improve the quality of gaseous and liquid products [22,23,24,25].

Gasification is a thermochemical process performed in presence of a sub-stoichiometric amount of oxygen with respect to that required for the complete combustion of a feedstock [26]. The gasification process transforms an initial matrix into a gaseous energy carrier called a syngas, consisting mainly of H_2_, CO, CH_4_ and CO_2_ [27,28]. However, this gas can be contaminated by inorganic and organic impurities such as solid particles, tars, sulfur and chlorine-containing compounds, etc. [29]. More in detail, tar is a black-brown viscous liquid consisting of a mixture of high-molecular-weight hydrocarbons which can lead to machinery malfunctions and clogging [30]. Different operating conditions are available for a gasification process: pressure can vary between 1 and 30 bar, the temperature can be set between 1000 and 1700 K, air, O_2_ and steam can be used as gasifying agent and the gasifier can be a fixed bed, a fluidized bed, a rotary kiln, etc. [28]. The quality of the gas is significantly affected by the gasification technology and the conditions applied. Typically, air-based gasifiers produce a syngas with a high nitrogen concentration and a lower heating value (4–7 MJ Nm^−3^), while O_2_/steam-based gasifiers produce a syngas with a high concentration of H_2_ and CO and a LHV in the range of 10–20 MJ Nm^−3^ [28].

Gasification is a promising technology for the conversion of mixed solid waste into a valuable gas to be further used in power generation systems [31,32]. Batch and semibatch gasifiers [33], fixed-bed gasifiers [34], fluidized bed gasifiers [35], entrained-flow gasifiers [28], plasma-assisted gasifiers [36] from the lab to pilot scale have been employed in the gasification of wastes, but a diffuse commercialization of such technologies still represent a challenge and only some of these plants (in Finland and Japan) are operating in the world. The high tar content in syngas is one of the most important disadvantages of the gasification process. The yield and characteristics of the tars depend on the feed composition and on the gasification technology used. Further reduction of tar can be achieved by physical or chemical ex situ methods such as scrubbing or secondary catalytic cracking [37]. The interest towards pyrolysis or gasification as thermoconversion processes is not only related to the possibility to recover energy and fuels from RDFs, but also to recover added-value materials suitable for construction industry applications. Indeed, gasification and pyrolysis products (liquid and solids products) can be used as additives in the pavement industry, reducing the need of natural resources or petroleum derivatives [4].

Despite the great interest in this field, the literature on the thermochemical treatment of a real RDF under pyrolysis and gasification conditions is still limited and a comprehensive characterization of the resulting products is rarely accomplished. In this paper, the results of pyrolysis and gasification tests on a real RDF in lab-scale reactors are reported. The performances obtained under the two different thermochemical processes (pyrolysis and gasification) are analyzed in terms of the yield, characteristics and quality of the products to highlight how the two thermochemical conversion processes can be used to accomplish waste-to-materials and waste-to-energy targets. In particular, pyrolysis tests have been performed in a fixed tubular reactor at three different temperatures (550, 650 and 750 °C) while the gasification test at 850 °C has been performed in a fluidized bed reactor to ensure extensive mixing between reagents and to keep a uniform process temperature with the aim of limiting the tar production.

The results herein disclose the aim to enrich the literature on the use of RDF as feedstock in thermochemical processes for the recovery of materials exploitable in the pavement industry. The circular-economy-based approach envisioned by this work is based on the exploitation of thermochemical processes to treat MSWs (or its fractions as Refuse-Derived Fuels, RDFs), whose residues (oil and char) can be used as additives in bitumen and asphalt preparation [4]. In particular, the proposed approach is aimed to pursue different possible benefits: (i) replacement of petroleum-derived products (e.g., crude oil) with products from the transformation of urban solid wastes; (ii) improvement of the mechanical characteristics and the longevity of asphalts; (iii) rejuvenation of exhausted asphalts. These benefits are expected to greatly impact aged asphalts disposal in landfills, wastes treatment, CO_2_ emission and production costs as a consequence of the increased asphalts duration.

## 2. Results and Discussion

### 2.1. Feedstock Compositional Characteristics

The results of proximate, ultimate and calorimetric analyses on the RDF are reported in Table 1.

The presence of a high-volatiles content (above 80% *w/w*) and a high-carbon content (48% *w/w*) is ascribable to the high content of plastics and paper-based materials [38]. The content of ashes is not negligible (9% *w/w*) and it is in line with other RDF compositions [38,39,40]. The ash content is an important parameter since it affects not only the calorific value and the amount of residue that would be left behind upon feedstock thermoconversion, but it can influence the feedstock reactivity under both pyrolysis and gasification conditions on the basis of its amount and composition [25]. The composition of ashes was determined by ICP-MS and the results are listed in Appendix A. The ashes consist primarily of alkali and alkali-earth-metal-containing minerals, with calcium-based compounds being the most abundant species. This aspect is particularly relevant, since alkali and alkali-earth-metal-containing minerals are expected to influence the thermal behavior of cellulose-based RDF components [20], while the thermal behavior of plastic components remains unaffected [41].

The low-heating value (LHV. 22.20 MJ/Kg) is in line with those reported for other real RDFs [35,40,42,43], higher with respect to the average value usually reported for MSWs (below 15 MJ/kg [6,44]) and those evaluated for other kinds of wastes applying the same conditions reported in this work (sewage sludge 10.6 MJ/Kg [45], paper sludge 11.1 MJ/kg [46], olive stone [47], lignin sludge 18.1 MJ/kg [48]) but lower than that of a plastic waste (>40 MJ/kg [35,49]).

### 2.2. Thermogravimetric Analysis

Figure 1 shows an RDF thermogravimetric profile and its corresponding derivative (DTG) line.

RDF decomposes through different thermal events: a first mass loss around 105 °C, ascribable to moisture removal, a second and intense mass loss peaked around 330 °C, ascribable to the decomposition of cellulose-based materials and soft plastic materials, a third and less intense mass loss peaked around 450 °C, ascribable to the decomposition of plastic-based components and a final mass loss around 700 °C, ascribable to the decomposition of carbonates [17,43,50,51]. Above 750 °C, the main events characterizing the thermal decomposition of RDF can be considered almost complete. However, it is important to underline that, after the third weight loss, the mass of the feedstock continues gradually to decrease up to the last weight loss, probably as consequence of the presence of the lignin component (from wood, accounting 2.5% of the whole neat RDF) and some more thermally resistant plastic components [15,17,50]. The overall thermal degradation profile of RDF is mainly determined by the thermal decomposition of its two most abundant fractions (cellulose- and plastic-based components); this behavior is in accordance with previous literature findings on the thermogravimetric behavior of RDF samples [15,39,50,52].

On the basis of the TG curve, 550 °C, 650 °C and 750 °C were selected as final temperatures to be set in the pyrolysis tests on a laboratory scale plant in order to evaluate the influence of final temperature on the product characteristics and to evaluate the possibility to orient the process to the production of a target molecule or a class of molecules.

### 2.3. Pyrolysis Tests Results

The outputs of the pyrolysis process are: a gaseous mixture, a carbon-rich solid (char) and a mixture of condensable species (water and organic compounds in liquid or solid form at room temperature). Table 2 reports the yields of these pyrolysis products. The yields have been determined as the weight ratio between the obtained products and the amount of feedstock loaded into the reactor.

The total yield is far from 100% in all the tests. In can be evidenced that the higher the final pyrolysis temperature, the worse is the mass balance closure, suggesting that this mismatch can be mainly ascribed to the condensation stage and the poor quantification of the gaseous fraction. Since an underestimation of the total gas production can occur due to the discontinuous analysis of some gaseous species, the calculated gas yield (as complement to 100 with respect to the other two products) is also reported.

Due to the intrinsic high-RDF composition variability and the use of different plant configurations, a reliable comparison with previous published data is not easy to accomplish. In addition, the frequent use of model RDFs instead of real RDFs makes this task more challenging. Keeping in mind this premise, the results here reported for the highest pyrolysis temperature (750 °C) are in good agreement with those reported by Efika and coworkers [39] and Bauh et al. [53]. In the study of Efika et al. [39] a real RDF with a C content around 50 wt.% was pyrolyzed in a tubular furnace up to 800 °C, obtaining, similarly to the present study, that the product with the lowest yield is the gaseous fraction and the product with the highest yield is the fraction made by condensable species [39]. In Bauh et al. [53], a real RDF characterized by a C content of around 40% and a volatile fraction around 64 wt.% was pyrolized in a fixed-bed reactor up to 700 °C, and the same trend in the yields of the pyrolysis products was detected.

#### 2.3.1. Characterization of Gaseous Fractions

Figure 2 reports the gas release profiles of H_2_, CO, CO_2_ and CH_4_ registered by online monitoring during the three pyrolysis tests.

The production of CO_2_ and CO start earlier with respect to those of CH_4_ and H_2_, reaching a maximum in correspondence of a temperature around 350 °C, namely next to the temperature of the first main decomposition event highlighted by the thermogravimetric analysis (Figure 1). Only in the case of the pyrolysis test up to 750 °C, additional CO_2_ and CO releases are detected above 700 °C, and the CO_2_ production probably is a consequence of carbonate decomposition while the CO release can be ascribed to secondary reactions, including auto-thermal gasification, Boudouard and methane dry reforming reactions [39]. The occurrence of some of these reactions can also explain the comparable CH_4_ formation at 650 and 750 °C and a higher H_2_ production during the pyrolysis test up to 750 °C.

The concentration profile has been integrated and the total amount of produced H_2_, CO, CO_2_ and CH_4_ calculated; Figure 3 reports the amount of H_2_, CO, CO_2_ and CH_4_ as a function of the final pyrolysis temperature. The production of CO_2_, CO and H_2_ increases with the increase of temperature, while the production of CH_4_ increases, moving from 550 °C to 650 °C, and remains constant above 650 °C. The yield of CO_2_ is the highest among all the detected gases, and this result is in line with other reports on the composition of the gaseous fraction deriving from the RDF pyrolysis [38,39]. The increase of CO and CH_4_ yields is lower than that observed for CO_2_ and H_2_, suggesting that the temperature increasing promotes methane steam reforming (CH_4_+H_2_O = CO_2_+H_2_) and the water gas shift reaction (CO+H_2_O = CO_2_+H_2_).

The amounts of the other hydrocarbon species, evaluated by micro-GC analysis are listed in Table 3.

As concerns the production of C2–C6 hydrocarbons, an increase is found for most of the species listed in Table 4 moving from 550 °C to 650 °C and a decrease moving from 650 °C to 750 °C; such a trend can be explained considering that high temperatures and inorganic species present in the RDF are expected to promote cracking reactions. Among the different C2–C6 hydrocarbons detected, ethylene (C_2_H_4_) was the most abundant; such a result was not surprising, considering the high amount of plastic components in the RDF under analysis, and agrees with the results reported by Zajemska et al. on an RDF rich in plastic components [50].

#### 2.3.2. Characterization of Solid Fractions

The char recovered from the pyrolysis plant after each pyrolysis test was analyzed by ultimate analysis. Carbon, hydrogen, nitrogen and sulfur contents are listed in Table 4.

As expected, the content of H decreases with the increase of the final pyrolysis temperature, since the decomposition and charring reactions lead to the release of hydrogen-containing molecules (charring reactions lead to the release of H_2_ and CH_4_ as volatile species [20,22,54,55]). The content of C is comparable in char@550 and char@650, and it decreases in char@750, probably as consequence of reactions occurring at temperatures above 700 °C, leading to carbon consumption [39]. The C content in the char samples is in line with that reported by previous authors [53,56] for other RDF samples, but it is lower compared to that of the char obtained after the pyrolyzation of lignocellulosic biomasses and pure cellulose [54,57] or plastic materials [58].

The char@550 °C is rich in N, suggesting that working at lower temperature allows to concentrate the N in the solid, preventing the release of N-containing compounds in the gas and liquid phases. This result is in line with the study proposed by Li et al. [59], where the release of nitrogen compounds during the pyrolysis of corn straw, lignite coal and a fuel obtained by their mixing was investigated using a horizontal tube furnace operated at a temperature range of 300−900 °C [59]. It can be also speculated that the content of S seems to increase with the temperature increase. This trend is not in accordance with [59]. This mismatch could be due both to the different feedstocks used by Li et al. and to the heterogeneity of our feedstock.

The presence of heteroatoms such as N and S into the carbonaceous matrix can be attractive for different applications: as catalyst, as adsorbent, as material for the development of components for energy harvesting modules and as additive for the pavement industry [4,22,60,61,62]. Biochar has been tested by different authors as a possible carbon-based low-cost additive in asphalt preparation [63,64]. These studies highlighted that the chemicophysical and morphological characteristics of biochar play a relevant role, since they are proven to interact with the macromolecules of the bitumen. On the basis of this observation, the possibility to tune the biochar surface chemistry and overall characteristics by simply changing the pyrolysis conditions is a topic of growing interest.

Morphological details of the three char samples were achieved by SEM imaging. Figure 4 reports the SEM images at different magnifications of the three char samples: an irregular fiber-like shape with a lumpy surface was highlighted for each sample. The presence of a fiber-like morphology is the result of decomposition mechanisms involving cellulose-based components such as paper, biomass residues and textiles [11]. The temperature seems to not strongly affect the morphology of the different chars, even if it is worth noting the presence of spherical particles rich in Ca, Si and alkali metals in char@750, as established by EDX analysis (Appendix A).

The thermal behavior of the three char samples in oxidative atmosphere has been evaluated by TG analysis in air (Figure 5, left panel). The TG profile of the RDF analyzed under the same conditions is reported for comparison. The RDF decomposes through a main mass loss peaking at 300 °C, small overlapping events around 450 °C and a final mass loss after 660 °C, leaving a residue around 20 wt.%. The TG profiles of all char samples are characterized by a main mass loss peaking around 420 °C due to the overlapping of different thermal events linked to the burn-off of the carbon-based matrix and a little mass loss above 680 °C ascribable to carbonate decomposition. This thermal behavior under oxidative conditions is in agreement with that reported by Lu et al. [56] for an RDF-derived char. The residue amount increases with the increase of the pyrolysis final temperature; indeed, the residue left after char@550 combustion is around 45 wt.% while that estimated at the end of char@750 combustion is above 60 wt.%.

The surface chemistry of the three char samples has been investigated by infrared spectroscopy; the FTIR spectra of the three char samples are reported as height-normalized spectra in Figure 5 (right panel) and shifted for clarity. The FT-IR spectrum of the RDF is characterized by a broad band around 3500 cm^−1^ ascribable to adsorbed water (moisture), a broad band around 2900 cm^−1^ due to the stretching modes of the C-H bonds [65], overlapping bands between 1700 and 900 cm^−1^ due to the single and double bonds involving C, H, N and O [65], a sharp peak at 875 cm^−1^ ascribable to CaCO_3_ and overlapping bands between 800 and 500 cm^−1^ due to the bending modes of aromatic C-H out-of-plane bonds [66]. This spectroscopic fine structure feature of the RDF FTIR spectrum is lost as consequence of the pyrolysis process; indeed, just after the pyrolysis at 550 °C, the overlapping bands around 2900 cm^−1^ disappeared, as well as those between 800 and 500 cm^−1^, while the overlapping bands between 1700 and 900 cm^−1^ became less structured and a peak centered at 1400 cm^−1^ became predominant. It is worth of noting that the peak ascribable to CaCO_3_ is still present after the pyrolysis process performed up to 750 °C.

#### 2.3.3. Characterization of Condensable Fractions

The condensable fraction recovered after each pyrolysis test comprises a waxy solid stuck on the walls of the reactor (wax-reactor@T) and of the flasks (wax-F1@T and wax-F2@T) and a condensed liquid recovered into the flasks (CS-F1@T and CS-F2@T). Waxes are reported as a typical pyrolysis product of MSW, RDF and plastics [67,68] and are exploitable in bitumen preparation, as they are proven to act as visco-elastic modulators of the bitumen [69].

The compositions of the waxy-like solids have been achieved by ultimate analysis and the results are listed in Table 5.

The amount of N in the waxes increases with the temperature increase; this result is in agreement with the N content decrease observed for the char samples (Table 4), confirming that the confinement of N in the char can be achieved only at lower temperatures. The C/H molar ratio is only slightly affected by the operative temperature and it is generally higher in the waxes recovered in the low-temperature-condensation stage (F2).

The condensed liquids collected in flask 1 and flask 2 have been analyzed by GC-MS as DCM solutions; the resulting TIC chromatograms are reported in Figure 6 and the complete list of identified compounds is reported in Appendix A. The listed compounds have been identified by comparison with a NIST library. In Appendix A, the reported relative abundances have been calculated as the ratio between the area of the single peak and the total area of the chromatogram.

As can be seen from the compounds listed in Appendix A, several species (oxygenated and nonoxygenated species) typical of the pyrolysis of lignocellulosic biomass and plastic components can be identified in CS -F1@T and CS-F2@T samples: short-chain carboxylic acids, furans, phenols, saturated and unsaturated aliphatic chains, aromatic compounds (also containing heteroatoms), nitro and other N-containing compounds (e.g., amine, pyridine derivatives), chlorinated compounds and polycyclic aromatic hydrocarbons (PAH). A selection of compounds representative of typical products from biomass and plastics pyrolysis has been made and their abundances compared in Figure 7 to highlight how their production depends on the temperature of the pyrolysis process and how they distribute in the different condensation flasks.

The low-molecular-weight compounds selected and reported in Figure 7 (acetic acid (AA), 2-furan methanol (F-OH), phenol (Ph-OH) and benzoic acid (BA)) are mostly present in flask 2, and this is not a surprising result, since flask 2 is located downstream of flask 1 and maintained at a lower temperature (−12 °C) with respect to flask 1. It is noteworthy that benzoic acid is one of the most abundant products in the CS mixtures and that its concentration shows at first an increase and then a decrease as the pyrolysis temperature increases. Unsaturated aliphatic chains show a higher abundance with respect to the saturated ones, which is in agreement with the C/H molar ratio obtained by elemental analysis (Table 5), and also cover a broad molecular-weight range. Finally, the content of aromatic derivatives and PAH shows an increase with the increase of the process temperature, suggesting once again that the temperature can be used to force the production of a specific class of compounds. These results agree with those reported by Efika et al. [39].

The waxy solids were also characterized by TG under an inert atmosphere (nitrogen, 40 mL/min) from 30 °C to 800 °C (HR 10 °C/min). The TG profiles and the corresponding DTG curves are reported in Figure 8. As expected, the waxy material stuck on the walls of flask 2 (located downstream of flask 1 along the condensation train) devolatilizes earlier compared to those recovered from the walls of the reactor and flask 1: this is mainly due to the lower average MW of the components of wax-F2@T. The waxy materials recovered from the walls of the reactor and flask 1 appear quite similar: the devolatilization rate of both materials reaches a maximum around 380 °C and the carbon residue is also quite comparable. Overall, the waxy material from flask 1 devolatilizes over a larger temperature range (50–500 °C) compared to that recovered from the reactor walls (200–500 °C), indicating the presence of a lighter fraction (lower MW) of waxes escaping the reactor. As regards the differences arising from the final pyrolysis temperatures, at 550 °C, the components of the waxy solid appear overall lighter compared to the materials recovered after conducting the pyrolysis at 750 °C.

The possibility to recover a condensable fraction reach in waxes makes the RDF a feedstock of choice for the production of additives for the pavement industry [4,69,70]. In particular, Abdy et al. in their review [69] stated that the use of plastic-based waxes in hot or warm mix technologies and in recycled-asphalt applications could potentially be a feasible solution to overcome the current limitations associated with raw plastic modifiers.

### 2.4. Gasification Test Results

The main product generated by the gasification process is a syngas (85.6 wt.%), but by-products as condensable species (1.72 wt.%, hereinafter tar) and char (around 12 wt.%, which is mainly composed by ashes of mineral origin) were also detected.

The syngas produced by the RDF gasification process under the operating conditions described in the experimental section and analyzed downstream of the condensation/filtering stages exhibited the composition reported in Table 6. As expected, the main components were carbon monoxide (31.31%), hydrogen (15.96%), methane (12.97%) and carbon dioxide (27.18%). The presence of C2-C4 hydrocarbons was ascribed to the decomposition of plastic materials present in the starting feedstock, in accordance with previous reports on MSW thermochemical conversion [50,71].

A volumetric ratio H_2_/CO = 0.51 was estimated. This value resulted lower than that typically found in industrial processes (H_2_/CO = ~1), suggesting that it would be advisable to add a further stage of the water gas shift (CO + H_2_O ↔ CO_2_+ H_2_) to improve the H_2_/CO ratio. Different authors, indeed, proposed the use of a two-stage process implementing a catalytic or a reformer stage after the gasification one, thereby also obtaining a considerable abatement of tar productions [32,37,72,73]. Ferreira et al. proposed the use of limestone inside the silica-based bed to minimize the production of pollutants [71].

The tar amount produced by RDF gasification was determined by weighing the condensable species collected during the sampling of the gas in analogy with the recovery of the condensation products of the pyrolysis tests. The weight of the tar collected during the sampling time is divided for the total volume of the gas sampled to obtain the tar concentration. The amount of water in the condensable species was determined by Karl–Fischer titration. The tar amount was evaluated by subtracting the water from the total amount of the condensable species. Tar characteristics are resumed in Table 7.

The tar concentration (3.15 g/m^3^), although far from the limits required for syngas applications such as internal combustion engines (<100 mg/Nm^3^) [74], is in line with the typical range reported for biomass gasification processes performed by using this reactor configuration (1–20 g/m^3^) [28]. The very low content of carbon and the high content of hydrogen measured by elemental analysis (referred to the whole condensed species) reflects the high content of water estimated by Karl–Fisher titration (about 80%).

The composition of syngas obtained in this work is in line with that reported (on free N_2_ basis) for a real RDF under similar experimental conditions by Arena et al. using an olivine-fluidized bed [35] or a pilot scale bubbling-fluidized bed reactor [75]. Some differences can be found in the tar concentration [33,75]. Arena and coworkers in both cases reported a very high tar concentration (54 g/Nm^3^ in [35] and 39 g/Nm^3^ in [75]), while in this work, a concentration of 3.15 g/Nm^3^ was obtained; it should be evidenced that the tar sampling protocols used are different and an overestimation of the tar concentration in the works of Arena et al. can be taken into account.

The condensed species were collected by washing the inner walls of the flasks of the condensing train (flask 1 (F1) at room temperature and flask 2 (F2) at −12 °C) with acetone (instead of DCM, due to the massive water presence with respect to the condensed phase recovered from the pyrolysis tests). The corresponding solutions (hereafter tar-F1 and tar-F2) were then analyzed by GC-MS (Figure 9) and their composition was determined by comparison with NIST libraries available in the data analysis software.

The chromatograms of the tar collected from the flask at room temperature (F1) and flasks at −12 °C (F2) are quite similar. Appendix A report the species identified in the two tar mixtures in order to highlight the complexity and variety of the condensable species produced. The relative abundances (Area%) are also reported in the tables, calculated by comparing the area underlying the peak compared to that of the entire chromatogram. Most of the compounds identified are polycyclic aromatic hydrocarbons (PAHs) and their alkylated derivatives. Naphthalene, biphenyl, biphenylene or acenaphthylene, fluorene, anthracene or phenanthrene and 2-phenyl-naphthalene are the most abundant products identified. As concerns the species with a low molecular weight typical of biomass thermoconversion (pyrolysis), only some furan derivatives were identified [37]. The absence of oxygenated compounds is ascribable to the effect of temperature on the tar composition; indeed, Blanco et al. [37] found that the amount of oxygenated compounds decreases as the gasification temperature increases.

The compositions of the condensable species recovered after the gasification process are quite different from those collected in the pyrolysis process at a comparable temperature (CS-F1@750 and CS-F2@750). The gasification process leads to the formation of a condensed phase richer in heavy hydrocarbon species, especially PAH, almost absent in those collected during the pyrolysis process.

The char mostly remains inside the bed and only few aliquots are dragged out by the gases (here and in the following: elutriated char) into flask 1. The char collected into flask 1 was recovered and analyzed.

At first, the morphology of the char was evaluated by SEM imaging and reported in Figure 10 at different magnifications. As in the case of the char recovered from pyrolysis tests, an irregular fiber-like shape (testifying the presence of cellulose-based components such as paper, biomass residues and textile) with a granular surface is evidenced. EDX analysis indicated the presence of Ca, Si and alkali metals.

The composition of the elutriated char was: C, 10.5 wt.%/H, 1.22 wt.%/N, 0.19 wt.%. This composition was very different from that of the solid residue recovered after the pyrolysis tests (also at high temperature, Table 5). This difference was also highlighted by comparing the thermal behavior of the char elutriated during gasification with that of the char@750 obtained in the pyrolysis (Figure 11). The two materials differ in the resistance towards oxidation and in the solid residue amount. The elutriated char exhibits a higher temperature of burn-off (around 480 °C compared with that of char@750 at around 410 °C) while the solid residue amount is 80% in the case of the elutriated char and 40% in the case of char@750. Such differences are due to gasification reactions that make the char more compact and graphitized and thus resistant to oxidation.

The surface chemistry of both char samples presents some similarities, as demonstrated by comparing the FTIR spectra in Figure 11, where the elutriated char is reported together with char@750 and RDF for a more comprehensive comparison. The main difference with char@750 arises from the presence of the characteristic band attributable to the oxygen functional groups (peaked around 1700 cm^−1^ and due to carboxylic and carbonylic functionalities, coupled with the broad bands in the 3700–3200 cm^−1^ region due to the stretching of the O-H groups), due to the presence of a small amount of oxygen during the gasification process. The persistence of the sharp peak at 875 cm^−1^ can also be observed, which is ascribable to the presence of CaCO_3_.

## 3. Materials and Methods

### 3.1. Materials

The RDF (25 kg) was supplied by Calabra Maceri srl (Rende, CS, Italy) and stored in a dry place. The RDF presented a nonhomogeneous aspect (Appendix A) and it was composed by large pieces of about 3 cm. Its average composition was: 84% plastics, 5.4% paper and cardboard, 3% multilayer packaging material, 2.5% wood; 0.7% organic fraction; 0.6% ceramics; 0.5% metals, 0.4% textile, 0.1% rubber; 0.1% glass, 0.1% leather, 2.6% others. The RDF-raw (residue-derived fuel as-received) was dried for 1 week (RDF-dry) at a controlled temperature (~30 °C) in order to reduce its moisture content (<10%) before its use in the pyrolysis and gasification tests, and grinded to reduce its size (RDF-milled), obtaining a more homogenous sample representative of the initial material to be used in characterization tests, in which a low amount of the sample is needed (elemental analysis, thermogravimetry).

In order to verify that the drying and the grinding treatments did not modify the feedstock main characteristics, the three different samples have been characterized using a macrobalance LECO 701 (as after reported), which allows to load the large pieces of about 3 cm of the starting material. It is possible to see from Appendix A that the values of the volatiles and the fixed carbon obtained on a dry basis or on a dry ash-free basis are very close to each other, confirming that the feedstock is not modified by the pretreatments. For this reason, only the label RDF was used to identify the feedstock.

### 3.2. Pyrolysis Tests

Three different pyrolysis tests have been performed at the three different final temperatures (550 °C, 650 °C, 750 °C), applying a heating rate (HR) of 30 °C/min under a nitrogen flux (60 NL/h). Two replicates of the same experimental run have been performed. The lab-scale plant used for pyrolysis testing is depicted in Figure 12.

The pyrolysis lab-scale plant consists of a cylindric quartz reactor (internal diameter of 26 mm) located in a temperature-controlled furnace made up by a Kantal resistance wrapped by a high temperature (1000 °C) refractory ceramic mantel and of a condensing train made up by four Erlenmeyer flasks, one at room temperature (first condensing stage, named in the text as flask 1) and three flasks cooled at −12 °C (water/ice/NaCl cooling bath), referred in the text as flask 2.

For each test, about 3 g of the RDF was loaded into the reactor. The feedstock was preloaded in a basket of metal net (inox–stainless steel) before the introduction into the reactor to avoid the dragging of the lighter components of the feedstock and the clogging due to the possible melting of some feedstock components. The reactor was inserted into a furnace, and once the set point value was reached, the temperature was kept constant until no further gas production was detected.

The pyrolysis output consists of a solid residue (char), a mixture of condensable species (CS) and incondensable gases. The condensable species were collected in the four flasks, while the incondensable gases (CO_2_, CH_4_, CO, H_2_) exiting the condensing train were continuously analyzed by online gas analyzers (ABB Magnos for O_2_; ABB Uras 14 for CO and CO_2_, ABB Caldos for H_2_, Vario LUXX for CH_4_) and also collected in sampling bags for offline analyses. The sampling bags were analyzed by a micro-gas chromatograph (Agilent 3000 A) equipped with 4 columns (OV-1, Alumina, PLOT-U, MS5A) for the determination of light hydrocarbons (C1–C6) in addition to H_2_, CH_4_, CO and CO_2_. The characteristics and the overall analysis conditions are reported in the Appendix A, along with the detected species.

The amounts of CO_2_, CH_4_, CO and H_2_ obtained during the tests have been evaluated by the integration of the gas profiles vs. time, while the amount of the other gaseous species detected in the bags (Appendix A) has been calculated by multiplying the mean concentration of each gas (obtained averaging the results of the analysis of each sampling bag) by the gas flow rate, assuming that the releasing time of each species was the same of that of methane estimated by online analysis.

The amount of char recovered at the end of the pyrolysis test inside the cylindrical quartz reactor was determined by weighting each plant element (cylindrical quartz reactor, tubes, net and fittings) before and after each pyrolysis test. The solid (labeled in the text as char@T (T is the final pyrolysis temperature) was recovered and stored in a dry environment for further analysis.

The waxy residues stuck on the reactor wall and inside the connection tubes (Appendix A) were mechanically recovered and stored separately from the char. The waxy residue recovered from the wall of the reactor was labeled as wax-reactor@T, while the one recovered from the connection tubes and walls of the flasks was labeled as wax-Fx@T (T is the final pyrolysis temperature, and Fx is referred to flask 1 or flask 2). The total amount of condensable species was determined by weighting the flasks before and after the process. Dichloromethane (DCM) was used to recover the condensable species from the flasks. The condensable species trapped inside flask 1 were labeled as CS-F1@T, while those collected in the three cooled flaks (flask 2) were labeled as CS-F2@T.

### 3.3. Gasification Tests

The lab-scale experimental setup used for the gasification test is schematically shown in Figure 13.

The experimental rig consists of a fluidized reactor bed (cylindrical stainless-steel reactor with an internal diameter of 41 mm) inserted into a temperature-controlled furnace, and ancillary equipment for online monitoring and characterization of all system outputs.

The primary gas is fed from the bottom; it passes through a section acting as preheater (internal diameter of 41 mm and height of 600 mm) made in steel for high temperatures and filled with ceramic rings to favor the heat exchange between the gas and the surrounding heated surfaces and to make uniform the gaseous flow. The preheater section ends with a flange in which a seat has been made to house a series of metal meshes, which act both as a distributor of the fluidizing gas and as a support for the material making the bed. Above the preheater section, there is the fluidization column (internal diameter of 41 mm and height of 400 mm) made in steel for high temperatures. Both the preheater and the column are located in an oven made up by electric resistors placed inside two half-shells of low-density ceramic fiber able to generate a total power of 5 kW. A PID temperature controller ensures the control of the operating temperature by means of a type K thermocouple inserted into the bed just above the preheater section. A pressure transducer, located near the thermocouple ensures the detection of possible pressure drops of the bed. The thermal isolation of the system (ovens and column) is realized by using an insulating ceramic mantel covered by an external aluminum case. At the exit of the reactor, an aliquot of the gas stream was online analyzed by means of gas analyzers (ABB Magnos for O_2_; ABB Uras 14 for CO and CO_2_, ABB Caldos for H_2_, Vario LUXX for CH_4_), and another aliquot was sampled by means of a high-precision pump to quantify the tar amount (more details are reported in the following), and the remaining part was sent to the hood. Before entering the analyzers, the hot gases pass at first through a filter to remove the elutriated solids and then through a trap to remove the water present in the stream, preventing possible interferences during online analysis ascribable to the IR detector contamination.

To ensure a homogeneous and constant feedstock supply inside the gasification reactor, the RDF was pelletized by means of a pressing operation with a hydraulic press. Approximately 100 pellets of about 0.5 g (100 × 0.5 g = 50 g total) with a diameter around 1 cm and a thickness around 0.5–0.8 cm were prepared. RDF pellets were manually fed into the reactor from the top (one every 30 s), reproducing a semicontinuous feeding. In order to effectively calculate the fuel-feed rate, 3 sets of pellets were prepared (set1, set2, set3), each consisting of 32 pellets and for each set, the total weight (set1 = 17.2 g, set2 = 16.9 g, set3 = 16.9 g) and the time taken for its introduction into the reactor were considered. This procedure allows also to verify the reproducibility of the test (the average gas composition obtained analyzing the data related to each set of pellets shows a variation lower than 10%).

The experimental conditions adopted for the gasification test are shown in Table 8.

As the fuel feeding started, the online monitoring of the process parameters and the sampling of the process outputs also started. During the test, the percentages of CO, CO_2_, CH_4_ and H_2_ on a dry basis were achieved; recording the gas concentrations along the entire gasification test duration, the concentration profiles as a function of time were obtained. As reported before, to quantify the produced tars, a definite amount of the gas stream was sampled and sent to a train of condensers, which allowed the collection of condensable species. The condensation train consists in a series of seven flasks (250 mL): the first condensing flask (F1) was kept at room temperature, while the other six ( F2) were cooled at −12 °C (water/ice/NaCl cooling bath). The gases leaving the condensing train, deprived of the condensable components, similarly to what is reported for the pyrolysis tests, are sampled in bags and analyzed offline by means of a micro-gas chromatograph (Agilent 3000A micro-GC) to determine the concentration of light hydrocarbons (detection limit ppm). Further details on the instrument and the analysis conditions applied are available in Appendix A. The total amount of condensable species was determined by weighting the flasks before and after the process. The condensed species were recovered from the flasks using acetone and analyzed. The condensed species from the flask at room temperature were collected separately (tar-F1) from those of the flasks maintained at −12 °C (tar-F2). The solid elutriated (named in the text as elutriated char) and collected in flask 1 was recovered and stored for further analysis.

At the end of the gasification test, the fuel supply was interrupted to start the “carbon-load” phase: to this aim, a stream of air was fed into the reactor to carry out the combustion of the char accumulated in the bed and the profiles of the produced gases acquired (mainly CO and CO_2_) over time. Through the integration of such profiles, the char content accumulated in the bed was estimated. After the carbon load phase, the reactor was cooled at room temperature.

### 3.4. Analytical Techniques

Proximate analysis to determine feedstock humidity, volatile, ashes and fixed carbon contents by means gravimetric evaluations (both direct and indirect) was performed on a LECO 701 thermobalance according to the standard ASTM D7582-15. Each measurement was repeated 4 times. C, H, N and S contents were determined by ultimate analysis following ASTM D3176-15 and ASTM D4239 standards. C, H and N contents were determined by using a LECO 628 analyzer after EDTA calibration (measurements were performed in triplicates). Sulfur content was determined by a LECO CS 144 analyzer calibrated with high-content (vanadyl sulphate pentahydrate) and low-content (low sulfur coal Leco 502-681) sulfur reagents (measurements were performed in duplicate).

Low-heating value (LHV) was determined in accordance with the ASTM D5865 by burning an amount of feedstock in pure oxygen in a PARR 6200 calorimeter (measurements were performed in duplicate).

The thermal behavior of the feedstock and of the pyrolysis and gasification products (both condensable species and char) was investigated through thermogravimetric analyses on a STA6000 Perkin-Elmer in inert (N_2_, 40 mL/min) or oxidizing (air, 40 mL/min) atmospheres. These conditions allowed for the evaluation of the thermal reactivity of each sample both in the absence and in the presence of oxygen. The thermal ramp was set from 30 °C to 800 °C, and depending on the analyzed material, the heating rate varied between 10 °C/min and 50 °C/min. An amount of 5–20 mg of material was loaded in an alumina crucible for each measurement. The alumina crucible was previously treated in furnace at 950 °C to guarantee an accurate solid residue determination.

Qualitative (composition of each mixture) and quantitative (relative abundance of each species in a mixture) information about the composition of the condensed species was achieved by a gas chromatographic analysis implementing a mass spectrometer as detector (GC-MS system). Each liquid mixture was analyzed by GC-MS without any further derivatization. The samples dissolved in DCM or acetone were filtered on a PTFE filter (Millipore, 0.45 μm pore size and 47 mm membrane diameter) to remove water traces and undissolved particles before the analysis. An amount of 1 μL of each solution was analyzed by an Agilent GC-MS instrument (7890A/5972C) equipped with an Agilent DB-624 capillary column (30 m × 0.25 mm i.d., 1.40 μm film thickness) and using He as the carrier gas (1.0 mL/min). The mass spectrometer was operated in electron ionization mode and an m/z range from 30 to 400 was scanned. The oven temperature was programmed as follows: the starting oven T (45 °C) was held for 3 min, then it was raised to 235 °C at a heating rate of 3 °C/min and held for 50 min.

The water content of the condensed species was determined by Karl–Fisher titration (METTLER TOLEDO V20 instrument); for each pyrolysis liquid, the measurement was repeated thrice.

The content of major inorganic elements was determined by Inductively Coupled Plasma Mass Spectrometry (ICP/MS) using an Agilent 7500CE instrument in accordance with the US-EPA 3051 and US-EPA 3052 methods. To this aim, an amount of feedstock was suspended with distilled water and treated under microwave heating with concentrate HNO_3_ and H_2_O_2_ for 30 min. The digested sample, after filtration, was diluted with distilled water and analyzed with the ICP-MS system. The measurement was repeated thrice.

The morphology of the char samples was evaluated by scanning electron microscopy (SEM) imaging using a FEI Inspect microscope equipped with an EDS Oxford AZtecLiveLite probe and Xplore 30 detector for elemental analysis. The powdered samples were previously dried and sputter-coated with a thin layer of gold to avoid charging.

The surface chemistry of the char samples was investigated by infrared spectroscopy measuring FT-IR spectra in the 450–4000 cm^−1^ range on a Perkin–Elmer Frontier MIR spectrophotometer operated in transmittance mode. The spectra were acquired on KBr pellets (2 wt. %), collecting 8 scans and correcting the background noise.

## 4. Conclusions

Pyrolysis tests in a lab-scale tubular reactor up to 550, 650 and 750 °C and an air gasification test at 850 °C in a fluidized bed reactor have been performed and the results analyzed to highlight how the two processes can accomplish waste-to-materials and waste-to-energy targets.

Among the three products collected as the outputs of the pyrolysis process, the most abundant and also the most promising in terms of possible applications in the pavement industry is the fraction of condensable species, whose highest yield was achieved at 550 °C. The massive presence of waxes makes this fraction, when used as a whole and without fractionation, a potential candidate for the replacement of fossil-fuel-based additives in bitumen formulation and asphalt processing and rejuvenation. Different literature reports, indeed, have demonstrated that the hydrocarbons in bio-oil can enrich the poor fraction of the maltene phase in exhaust asphalts, promoting a rejuvenation process.

It is worth of noting also that the final pyrolysis temperature has a strong influence on the segregation of some critical species such as S and N in the char, opening the door to its reuse as adsorbent, catalyst, material for energy harvesting devices and additives for the pavement industry, depending on its composition. Char is mainly made up by carbonaceous particles and it is highly compatible with the organic nature of bitumen; as a consequence, the char addition is expected to reinforce the overall bitumen structure, increasing its mechanical properties and slowing down the molecular kinetics of its aging process. On the basis of these expectations, char is a low-cost carbon-based additive candidate for next-generation bitumen formulation.

The gasification process leads to the production of a syngas with a H_2_/CO ratio of 0.51. The result is in line with other literature reports, as well as those regarding pilot-scale systems, but to obtain a marketable product, a two-stage process implementing a catalytic or a reformer stage after the gasification one, also to reduce tar production, should be envisioned.

To sum up, with this work, the achievement of a possible integration of urban wastes and asphalt cycles is envisioned accordingly with the paradigm of the circular economy: thermochemical processes are used treat MSWs (or its fractions as Refuse-Derived Fuels, RDFs) under specific conditions, allowing to tune the characteristics of the resulting outputs (oil and char), making them feasible additives for bitumen and asphalt preparations. Since char can be used to prepare better performing and durable asphalts, and oil can be used to regenerate exhausted asphalts, avoiding their landfilling, the expected benefits of the proposed approach influence asphalts disposal, wastes treatment, CO_2_ emissions and overall production costs as a consequence of the increased asphalt duration.

## Figures and Tables

**Figure 1 molecules-27-08114-f001:**
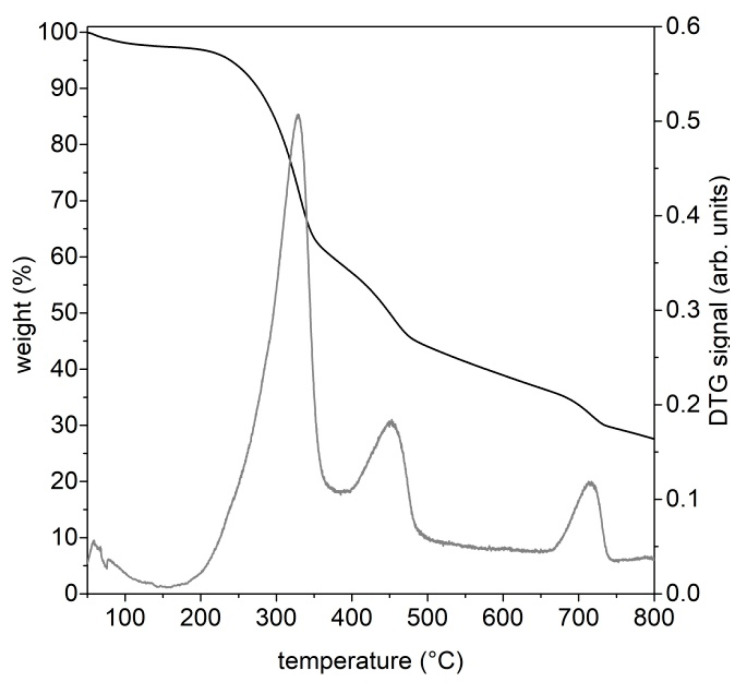
Thermogravimetric curve of RDF at 30 °C/min, N_2_ atmosphere (black line) and the corresponding derivative (DTG) curve (grey line).

**Figure 2 molecules-27-08114-f002:**
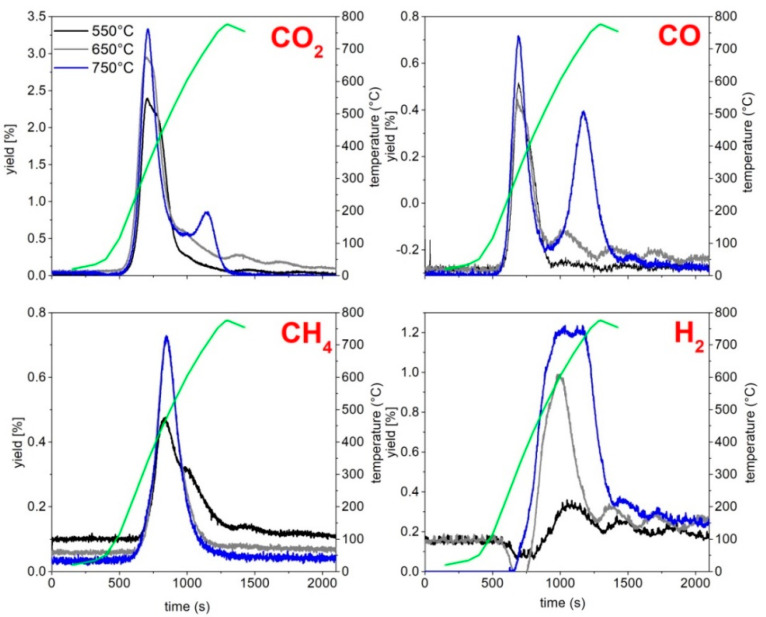
Gas release profiles of pyrolysis tests at 550 °C, 650 °C and 750 °C.

**Figure 3 molecules-27-08114-f003:**
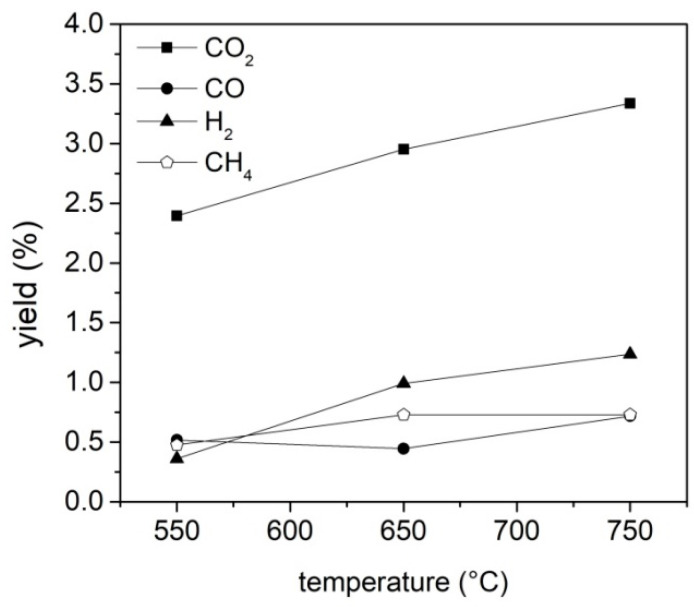
Yields of H_2_, CO, CO_2_ and CH_4_ as a function of the final pyrolysis temperature.

**Figure 4 molecules-27-08114-f004:**
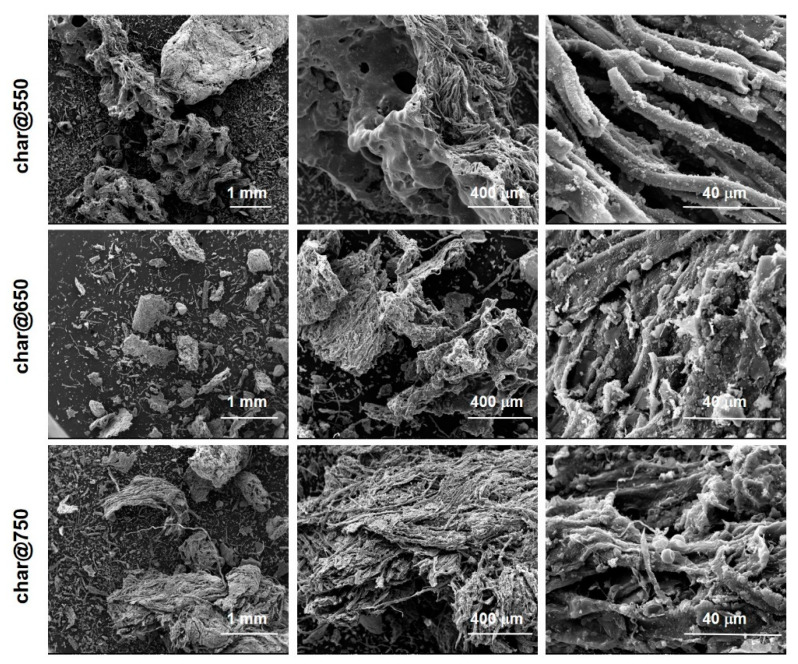
SEM images of char@550, char@650 and char@750 at different magnifications.

**Figure 5 molecules-27-08114-f005:**
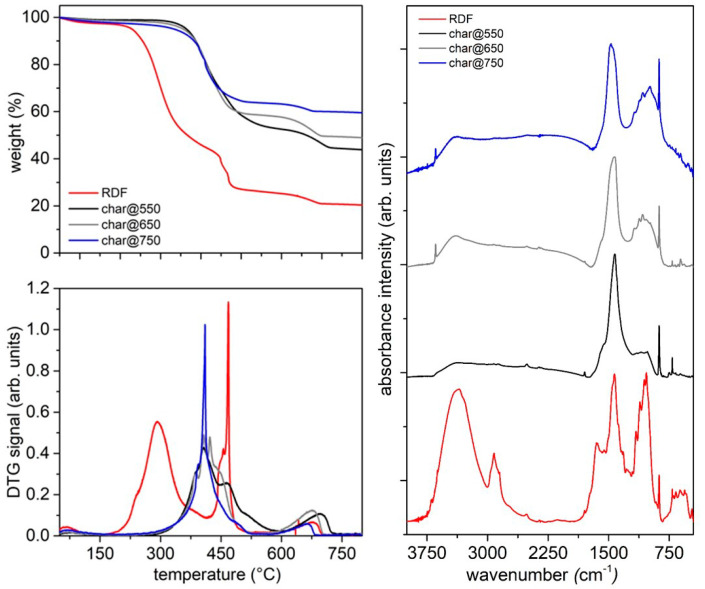
Left panels: TG profiles (up) and DTG curves (down) of char samples and RDF in oxidative atmosphere (synthetic air, flux 40 mL/min, HR 10 °C/min); right panel: height-normalized FTIR spectra of RDF, char@550, char@650 and char@750.

**Figure 6 molecules-27-08114-f006:**
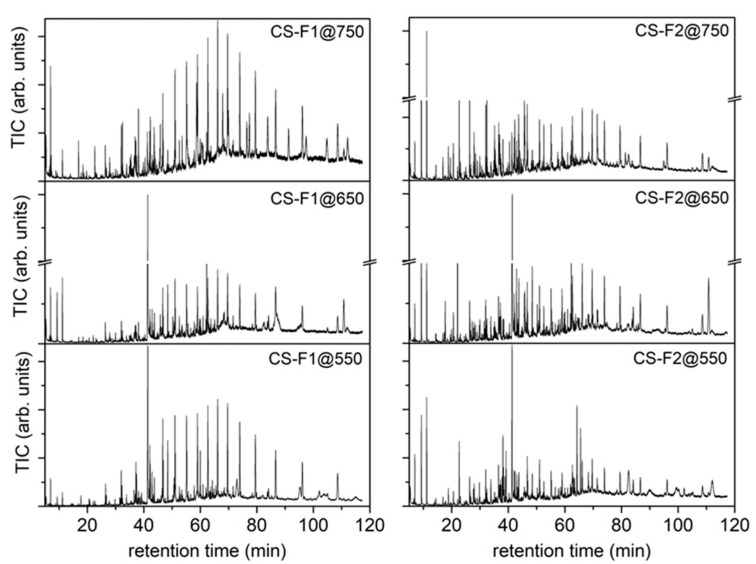
TIC chromatograms of the condensed species recovered from flask 1 and flask 2 after RDF pyrolysis tests up to 550, 650 and 750 °C.

**Figure 7 molecules-27-08114-f007:**
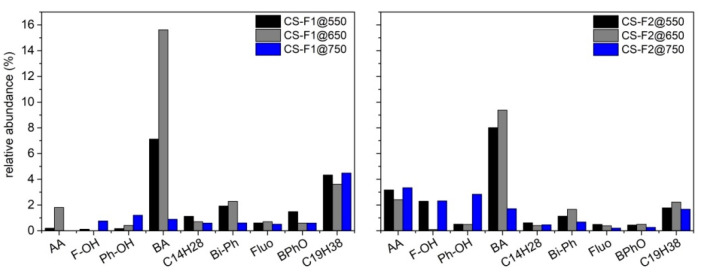
Comparison of the abundances of selected species present in CS mixtures collected in flask 1 and flask 2 after RDF pyrolysis tests up to 550, 650 and 750 °C (acetic acid (AA), 2-furan methanol (F-OH), phenol (Ph-OH), benzoic acid (BA), tetradecane (C_14_H_28_), biphenyl (Bi-Ph), fluorene (Fluo), benzophenone (BPhO) and 1-nonadecene (C_19_H_38_)).

**Figure 8 molecules-27-08114-f008:**
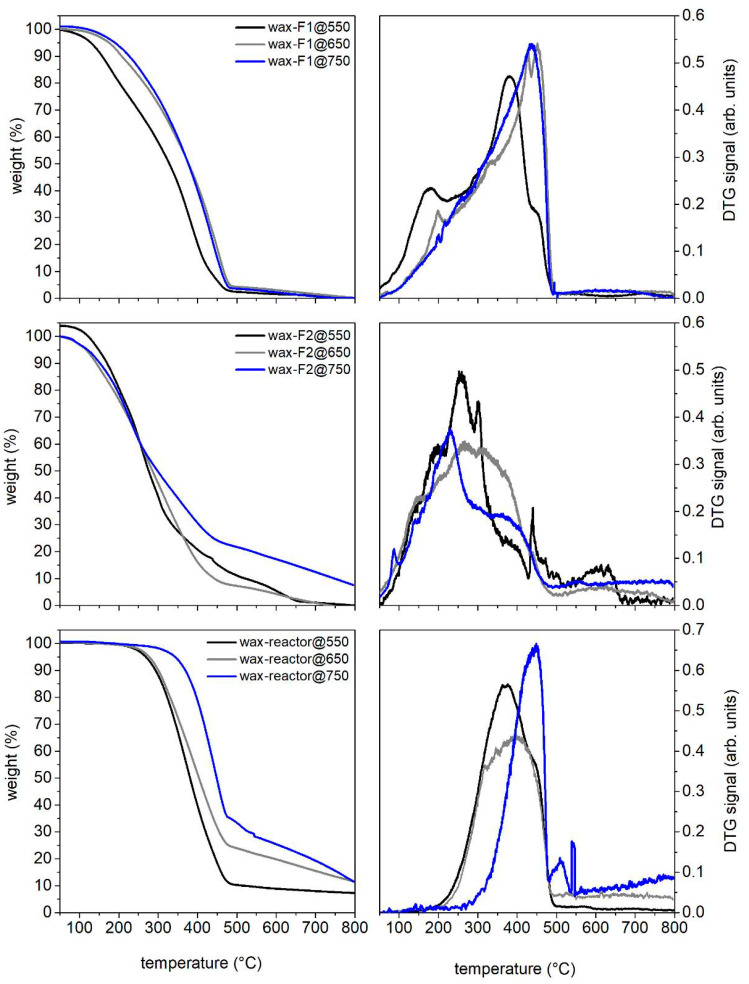
TG profiles and the corresponding DTG curves of waxy solids recovered after RDF pyrolysis tests up to 550, 650 and 750 °C. TG analyses have been performed under nitrogen atmosphere (flux 40 mL/min, HR 10 °C/min).

**Figure 9 molecules-27-08114-f009:**
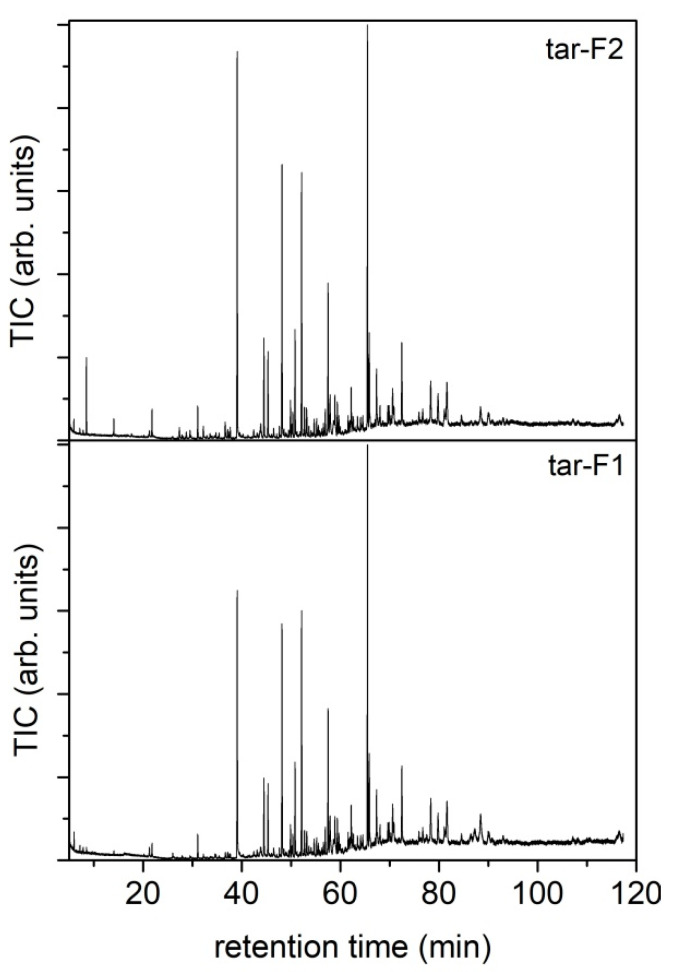
GC-MS chromatograms of tar-F1 and tar-F2.

**Figure 10 molecules-27-08114-f010:**
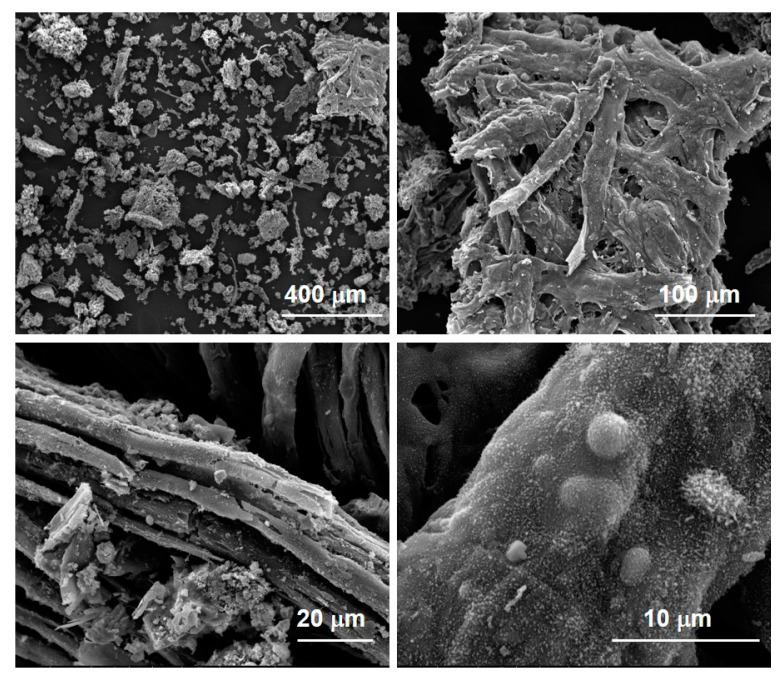
SEM images of elutriated char at different magnification.

**Figure 11 molecules-27-08114-f011:**
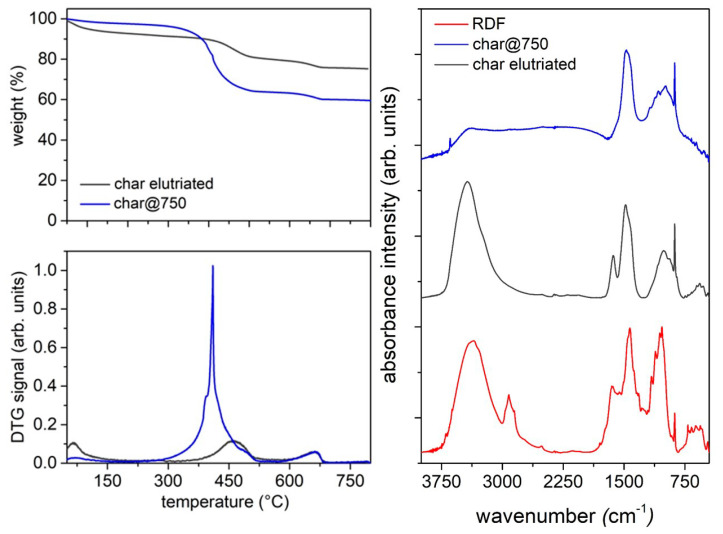
Left panels: TG curves (up) and the corresponding DTG curves (down) of the elutriated char and char@750 (oxidizing atmosphere, air); right panel: FTIR spectra of RDF, elutriated char and char@750.

**Figure 12 molecules-27-08114-f012:**
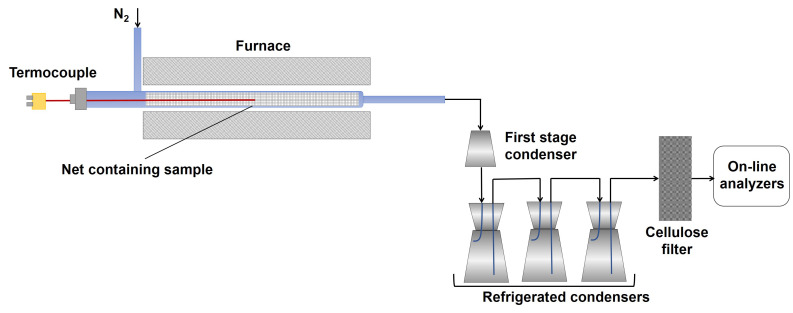
Lab-scale pyrolysis plant.

**Figure 13 molecules-27-08114-f013:**
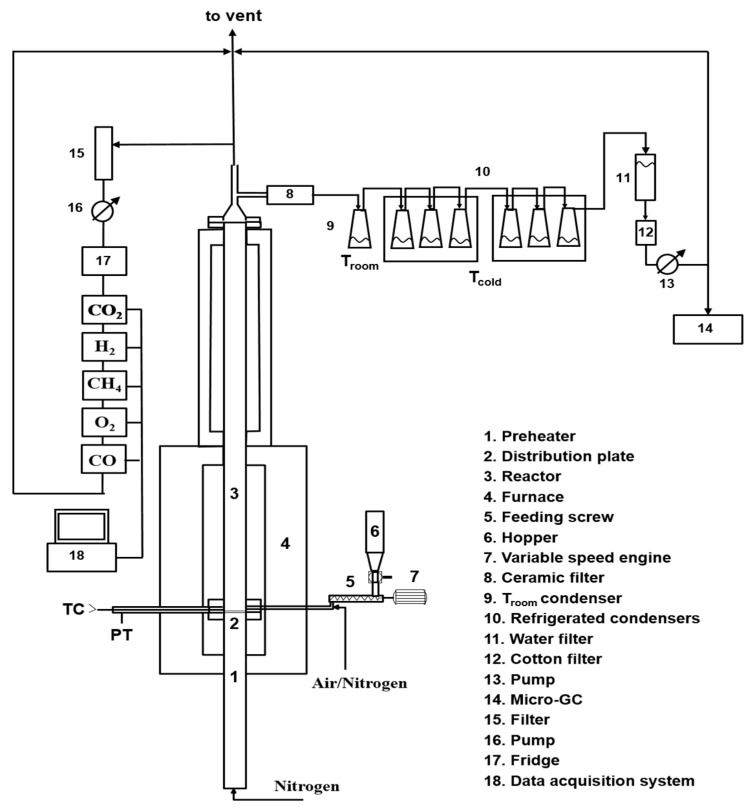
Lab-scale gasification test plant (adapted from [47]).

**Table 1 molecules-27-08114-t001:** RDF composition and properties.

Ultimate analysis	
C (%*w/w*)	48.39
H (%*w/w*)	6.85
N (%*w/w*)	0.39
S (%*w/w*)	0.30
Proximate analysis	
Humidity (%*w/w*)	1.61
Volatiles (%*w/w*)	80.7
Ashes (%*w/w*)	9.05
Fixed carbon (%*w/w*)	8.70
Calorimetric analysis	
Low heating value (MJ/kg)	22.20

**Table 2 molecules-27-08114-t002:** Products yields of the thre pyrolysis process on the RDF at the different reaction temperatures.

	550 °C	650 °C	750 °C
Char yield (%*w/w*)	23.9	21.36	27.95
Condensable species yield (%*w/w*)	51.96	45.81	31.75
Gas yield (%*w/w*)	10.73	16.12	10.61
Total yield	86.59	67.66	60.03
Calculated Gas yield (%*w/w*) *	24.14	32.83	40.30

* calculated as complement to 100 with respect to the other two products.

**Table 3 molecules-27-08114-t003:** Gaseous fractions composition estimated by micro-GC analysis. Values are reported as vol.%.

	550 °C	650 °C	750 °C
ethane (C_2_H_6_)	0.0416	0.0455	0.0287
n-propane (C_3_H_8_)	0.1014	0.0201	0.0029
isoC_4_	0.0001	0.0000	0.0002
isoC_5_	0.0000	0.0000	0.0004
isoC_6_	0.0012	0.0014	0.0003
nC_5_	0.0248	0.0222	0.0067
nC_6_	0.0033	0.0020	0.0021
ethylene (C_2_H_4_)	0.0707	0.0852	0.0371
n-butene (C_4_H_8_)	0.0558	0.0545	0.0284
n-pentene (C_5_H_10_)	0.0114	0.0117	0.0073
3-methyl-1-butene (C_5_H_10_)	0.0024	0.0045	0.0000
n-hexene (C_6_H_12_)	0.0202	0.0197	0.0110

**Table 4 molecules-27-08114-t004:** Composition of char samples from RDF pyrolysis up to 550, 650 and 750 °C.

% *w*/*w*	char@550	char@650	char@750
C	44.92	45.23	42.00
H	1.58	0.98	0.50
N	2.16	1.31	1.07
S	0.33	0.71	1.02

**Table 5 molecules-27-08114-t005:** Composition of waxy-like solids from RDF pyrolysis up to 550, 650 and 750 °C.

	C, wt.%	H, wt.%	N, wt.%	C/H Molar Ratio
wax-F1@550	80.07	12.20	0.11	0.55
wax-F1@650	71.48	10.85	0.65	0.55
wax-F1@750	79.90	13.05	0.17	0.51
wax-F2@550	72.24	9.68	1.09	0.62
wax-F2@650	62.01	7.37	0.61	0.70
wax-F2@750	57.69	8.68	0.79	0.55
wax-reactor@550	80.89	12.09	1.27	0.56
wax-reactor@650	74.54	9.14	1.25	0.68
wax-reactor@750	74.69	8.93	2.12	0.70

**Table 6 molecules-27-08114-t006:** Composition of the syn-gas produced by the RDF gasification. The percentages are reported on a water- and nitrogen-free basis.

%	Syn-Gas	Syn-Gas Free of N_2_
CO_2_	2.71	27.18
CO	3.12	31.31
H_2_	1.59	15.96
CH_4_	1.29	12.97
C_2_H_4_	0.4	4.01
C_2_H_6_	0.01	0.10
C_4_H_8_	0.06	0.60
C_3_H_6_	0.78	7.83
C_2_H_2_	0.001	0.01
C_5_H_10_	0.002	0.02
N_2_	90.0	-

**Table 7 molecules-27-08114-t007:** Condensed species (CS) characteristics.

Tar mass * (g)	0.302
Water mass (g)	1.36
Water percentage in CS (%)	81.8
Water volume (NL)	1.69
Sampled gas total volume (NL)	81.5
Tar concentration (g/m3)	3.15
Elemental composition *	C, 0.51wt%/H, 8.06 wt.%/N, 0.23 wt.%

* referred to the whole condensed species (water + tar).

**Table 8 molecules-27-08114-t008:** Operating conditions adopted during the gasification test.

Bed material	Sand
Bed material dimensions, mm	0.3–0.4
Bed quantity, kg	0.18
H/D Ratio	3
Minimum fluidization rate @T_bed_, m/s	0.044
Fed fuel	RDF-dry pellet
Fuel-feeding flow rate, g/h	64
Reaction temperature, °C	850
Inlet air flow rate, NL/h	50
Inlet N_2_ flow rate, NL/h	300
Gas total flow rate, NL/h	350
Superficial gas speed @T_bed_, m/s	0.3
Equivalence ratio	0.18

## Data Availability

Data is contained within the article or Appendix A.

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
