# Peer review of "Pyrolysis and Gasification of a Real Refuse-Derived Fuel (RDF): The Potential Use of the Products under a Circular Economy Vision"

_molecules, 2022, doi:10.3390/molecules27238114_

Round 1
Reviewer 1 Report
The manuscript is good and is about an important field in the current time. I have some clarifications and recommendations for the authors before taking up the manuscript forward.
It was mentioned that the fuel has High calorific value, which is not true compared to the conventional fuels used, which is around 45 MJ/kg. So I recommend that the authors rewrite this or make a suitable comparison to justify the calorific value is higher for their fuel.
“RDFs more suitable than MSWs for thermochemical valorization purposes”, aren’t RDFs derived from MSWs in the first place? So how do you validate this process?
What does the authors imply on the term Real RDF?
Table 2: It was mentioned that the calculated gas yield is use to complement the 100% balance. However, for the 550 C temperature the total was going way higher than 100%. Please explain this also.
I came across this manuscript, where the authors have condensed most of the gaseous content into liquid oil, which I believe reduced errors in their quantification process - https://doi.org/10.1016/j.jclepro.2021.127687. Also they have used two types of pyrolysis process. Has anything like that been tried by the authors?.
Page 8, line 280: char@550°C is rich in N and S suggesting that working at lower temperature 280 allows to concentrate the N and S in the solid preventing their release in the gas and liquid 281 phase. However it is very clear from the table 4, that sulphur concentration significantly increases with temperature. How does the authors justify this.
Page: 18: what kind of metal net was used, where they kept throughout the pyrolysis process?
Section 3 should be put as section 2 and vice versa. This will help the readers understand the process and setup properly before going into the results part.
The authors have mentioned the calorific value of the RDFs in the abstract, however, I am not able to find any such information in the present research manuscript. Then how was the calorific value measured? If it was from literatures, why was is mentioned in the abstract and also why the authors didn’t consider to do the testing.
The conclusion reports the process only and not about the final outcome of the work. Also how this helps in the circular economy is also not very clear.
Giving a final application point might be very useful in the conclusions.
Author Response
Reviewer#1
Comments and Suggestions for Authors
The manuscript is good and is about an important field in the current time. I have some clarifications and recommendations for the authors before taking up the manuscript forward.
We thank the Reviewer for his/her positive evaluation of our work.
Point 1: It was mentioned that the fuel has High calorific value, which is not true compared to the conventional fuels used, which is around 45 MJ/kg. So I recommend that the authors rewrite this or make a suitable comparison to justify the calorific value is higher for their fuel.
Reply: We thank the Reviewer for giving us the possibility to explain better this aspect related to the feedstock used in the pyrolysis and gasification tests. The Reviewer is right, conventional gaseous and liquid fuels are characterized by high calorific values (up to 45-50 MJ/kg), while a solid fuel exhibits generally lower calorific values (e.g. coals are characterized by LHV values not exciding 30 MJ/Kg as reported in the table reported in: https://www.engineeringtoolbox.com/fuels-higher-calorific-values-d_169.html).
For MSWs and RDFs, namely fuels which consist of a mixture of several different compounds, the calorific value varies as function of the composition. The RDF used in our tests is characterized by a LHV of 22.20 MJ/Kg, a value in line with those reported for other real RDFs (Akdag et al.2016; Danias et al. 2018), higher than reported for MSWs (Montejo et al. 2011) and those evaluated for other kinds of wastes applying the same conditions reported in this work (sewage sludge 10.6 MJ/Kg [Migliaccio et al. 2021], paper sludge 11.1 MJ/kg [Areeprasert et al. 2016], olive stone [Ganda et al. 2022], lignin sludge 18.1 MJ/kg [Miccio et al. 2016], but lower than that of plastic waste 40.4 MJ/kg [Arena et al. 2012]).
In accordance with the suggestion of the Reviewer, these details have been added to the revised manuscript.
Akdağa, A.S.; Atımtay, A.; Sanin, F.D. Comparison of fuel value and combustion characteristics of two different RDF samples. Waste Management 2016, 47, 217–224
Danias, P.; Liodakis, S. Characterization of Refuse Derived Fuel Using Thermogravimetric Analysis and Chemometric Techniques. J. Anal. Chem., 2018, 73(4), 351–357.
Montejo, C.; Costa, C.; Ramos, P.; del Carmen Márquez, M. Analysis and comparison of municipal solid waste and reject fraction as fuels for incineration plants. Appl. Therm. Eng. 2011, 31(13), 2135-2140.
Migliaccio, R.; Brachi, P.; Montagnaro, F.; Papa, S.; Tavano, A.; Montesarchio, P.; Ruoppolo, G.; Urciuolo, M. Sewage Sludge Gasification in a Fluidized Bed: Experimental Investigation and Modeling. Ind. Eng. Chem. Res. 2021, 60, 5034−5047
Areeprasert, C.; Scala, F.; Coppola, A.; Urciuolo, M.; Chirone, R.; Chanyavanich, P.; Yoshikawa, K. Fluidized bed co-combustion of hydrothermally treated paper sludge with two coals of different rank Fuel Processing Technology 144 (2016) 230–238
Ganda, E.T.; Brachi, P.; Urciuolo, M.; Migliaccio, R.; Coppola, A.; Scala, F.; Salatino P.; Ruoppolo, G. Catalytic pyrolysis of torrefied olive stone for production of potential petrochemical alternatives. Biofuel Bioprod Bior., 2022, 2397
Miccio, F.; Solimene, R.; Urciuolo, M.; Brachi, P.; Miccio, M. Fluidized Bed Combustion of a Lignin-based Slurry. Chem. Eng. Trans. 2016, 50, 271-276.
Arena U., Chirone R., Di Gregorio F., Solimene R., Urciuolo M., Zaccariello L. A Comparison Between Fluidized Bed Combustion and Gasification of a Mixed Plastic Waste (ISBN) 9788889677834 21st International Conference on Fluidized Bed Combustion 2012
Point 2: “RDFs more suitable than MSWs for thermochemical valorization purposes”, aren’t RDFs derived from MSWs in the first place? So how do you validate this process?
Reply: MSWs consists of three major fractions: a combustible fraction, a non-combustible fraction and moisture or volatile material. The combustible fraction is separated from the other two by specific sorting facilities [Rezaei et al. 2020]. MSWs, typically, are processed to remove the recyclable fraction (e.g. metals), the inert fractions (such as glass) and separate the fine wet organic fraction (e.g. food and garden waste) containing high moisture and high ash material. There are two basic sorting methodologies which have been developed to produce MSWs derived fuel [Psomopoulos et al. 2014]: Mechanical Biological Treatment (MBT) and Biological Drying Process. In a mechanical biological treatment plant (MBT), metals and inert materials are separated out and organic fractions are screened out for further stabilization using composting processes, either with or without a digestion phase. In a biological drying process, MSWs are effectively dried (and stabilized) through a composting process, leaving the residual mass with higher calorific value and suitable for combustion. The inert materials and metals are removed through mechanical process before or after the bio-drying depending of the technology applied for bio-drying [Psomopoulos et al. 2014]. Waste-derived fuels are usually referred as refuse-derived fuel (RDF) or solid recovered fuel (SRF), depending upon the fuel’s characteristics. SRF is more homogeneous and less contaminated than the generic RDF. RDF composition varies with season, location of waste collection and efficiency of MSW sorting. Average RDF composition can be described as follows: 15–35% plastic, 15–50% cellulosic paper and cardboard, 2–10% wood, 5–20% organics and about 5–10% non-combustible matter [Psomopoulos et al. 2014, Raizei et al. 2020]. Since the RDF production concept ensures a certain degree of size reduction and removal of organic and inert material, RDF is characterized, on average, by higher heating value, lower ash content, and a lower bulk density compared to untreated waste.
These details have been added to the revised version of the manuscript.
Rezaei, H.; Panah, F.Y.; Lim, C.J.; Sokhansanj, S. Pelletization of Refuse-Derived Fuel with Varying Compositions of Plastic, Paper, Organic and Wood. Sustainability 2020, 12, 4645; doi:10.3390/su12114645
Psomopoulos, C. S.. Residue derived fuels as an alternative fuel for the Hellenic power generation sector and their potential for emissions reduction. AIMS Energy 2014, 2(3), 321-341.
Point 3: What does the authors imply on the term Real RDF?
Reply: The feedstock used in our pyrolysis and gasification tests derived from a mechanical biological treatment plant and it is not a result of a mixing of components to simulate the average composition of a typical RDF. We underlined this aspect, because in the literature most of the works deal with simulated RDFs.
Point 4: Table 2: It was mentioned that the calculated gas yield is use to complement the 100% balance. However, for the 550 C temperature the total was going way higher than 100%. Please explain this also.
Reply: We thank the Reviewer for the advice. The Table 2 has been revised accordingly.
Point 5: I came across this manuscript, where the authors have condensed most of the gaseous content into liquid oil, which I believe reduced errors in their quantification process - https://doi.org/10.1016/j.jclepro.2021.127687. Also they have used two types of pyrolysis process. Has anything like that been tried by the authors?
Reply: The reviewer is right; the condensation stage is a critical step for the evaluation of the liquid fraction yield as it is associated to a high quantification error. The efficiency of condensation stage affects also the gas yield evaluation. As a general rule, more efficient is the condensation stage as lower the error in the yields estimation is. Different strategies can be adopted to improve the condensation stage, and the use of a circulating refrigerating system, as proposed in https://doi.org/10.1016/j.jclepro.2021.127687, is an interesting possibility. This configuration has not been tried jet in our set up, but studies to improve the condensation stage in our plant are ongoing.
Point 6: Page 8, line 280: char@550°C is rich in N and S suggesting that working at lower temperature 280 allows to concentrate the N and S in the solid preventing their release in the gas and liquid 281 phase. However it is very clear from the table 4, that sulphur concentration significantly increases with temperature. How does the authors justify this.
Reply: We thank the Reviewer for this valuable comment. The reviewer is right, only N seems to be concentrated in char at 550 °C. This trend is in accordance with the work of Li et al (2020).
Looking at the data listed in Table 4, it can be also speculated that the content of S seems to increase with the temperature increase. This trend is not in accordance with the above mentioned work of Li et al (2020). This mismatch could be due both to the different feedstocks used by Li et al. (corn straw and lignite coal) and to the heterogeneity of our feedstock. For the above reasons, the trend we observed needs to be consolidated in dedicated experimental campaigns.
In accordance with the suggestion of the Reviewer, these aspects have been added to the revised manuscript.
Li et al. ACS Omega 2020, 5, 30001−30010
Point 7: Page: 18: what kind of metal net was used, where they kept throughout the pyrolysis process?
Reply: The net used as seat for the feedstock during pyrolysis test is made of inox-stainless steel and it is inserted into the reactor (quartz tube) before starting the warm up of the plant. This detail has been added to the revised version of the manuscript.
Point 8: Section 3 should be put as section 2 and vice versa. This will help the readers understand the process and setup properly before going into the results part.
Reply: We agree with the Reviewer’s preference, but, unfortunately, the template provided by the Journal requires that the experimental section comes after the Results and Discussion section.
Point 9: The authors have mentioned the calorific value of the RDFs in the abstract, however, I am not able to find any such information in the present research manuscript. Then how was the calorific value measured? If it was from literatures, why was is mentioned in the abstract and also why the authors didn’t consider to do the testing.
Reply: The low heating value (LHV) of the RDF used in pyrolysis and gasification tests has been experimentally determined by a calorimetric analysis using a PARR6200 calorimeter in accordance with ASTM D5865 standard. The estimated value was 22.20 MJ/Kg and it is listed in table 1 along the other properties of RDF. For sake of clearness, Table 1 has been revised.
Point 10: The conclusion reports the process only and not about the final outcome of the work. Also how this helps in the circular economy is also not very clear.
Reply: The circular economy-based approach envisioned by this work is based on the use of thermochemical processes to treat MSW (or its fractions as Refuse Derived Fuels, RDFs), whose residues (oil and char) can be used as additives in bitumen and asphalt preparation. Char can be used to prepare better performing and durable asphalts, and oil can be used to regenerate exhaust asphalts, avoiding their landfilling since: (i) char is made up by carbonaceous particles highly compatible with the organic nature of bitumens (its addition is expected to reinforce the overall bitumen structure, increasing its mechanical properties and slowing down the molecular kinetics of its aging process); (ii) oil is rich in hydrocarbons, so it can enrich the poor fraction of the maltene phase in exhaust asphalts. With this work, we would like to stress how a possible integration of urban wastes and the asphalt cycles can be achieved. In particular, we would like to show how the thermoconversion product characteristics can be tuned by changing process conditions and how added-value materials with composition and properties suited for asphalt formulation and rejuvenation can be produced by a proper treatment of MSW (or its fractions as Refuse Derived Fuels, RDFs). The proposed approach is linked to also different possible benefits: i) replacement of petroleum-derived products (e.g., crude oil) with products from the transformation of urban solid wastes; ii) improvement of the mechanical characteristics and the longevity of asphalts; iii) rejuvenation of exhausted asphalts. The above-mentioned benefits are expected to greatly impact aged asphalts disposal in landfills, wastes treatment, CO2 emission, and production costs, as a consequence of the increased asphalts duration.
In accordance with the suggestion of the Reviewer, these aspects have been added to the revised manuscript to better frame the work in a circular economy perspective.
Point 11: Giving a final application point might be very useful in the conclusions.
Reply: We thank the Reviewer for the suggestion. Innovative applications of pyrolysis derived products in asphalt preparation have been proposed by different authors (e.g. bio-oil has been proposed as additive for asphalt preparation and rejuvenation agent of exhausted asphalts in substitution of petroleum-derived products, biochar has been proposed as low-cost carbon based reinforcement of bitumen structure to increasing its mechanical properties and slowing down the molecular kinetics of its aging process), and some recent review papers have been also published recently (Raman et al. 2020; Rondón-Quintana et al 2022). On the basis of this, the Conclusions section has been revised accordingly.
Raman, N. A. A., Hainin, M. R., Hassan, N. A., & Ani, F. N. (2015). A review on the application of bio-oil as an additive for asphalt. Jurnal Teknologi, 72(5).
Rondón-Quintana et al. Use of Biochar in Asphalts: Review. Sustainability 2022, 14, 4745. https://doi.org/10.3390/su14084745
Reviewer 2 Report
This is an interesting study about the reusability study of derived fuels waste toward circular economy. Whether it is really of significance to circular economy is not clear, but the practical application of derived fuel is meaningful.
Some comments
1) How about the actual cost of the developed method for derived fuel?
2) suggest to add some performance comparison with other similar studies in the field, especially regaridng the real engineering application
Author Response
Reviewer#2
Comments and Suggestions for Authors
This is an interesting study about the reusability study of derived fuels waste toward circular economy. Whether it is really of significance to circular economy is not clear, but the practical application of derived fuel is meaningful.
We thank the Reviewer for his/her positive evaluation of our work.
Some comments
Point 1: How about the actual cost of the developed method for derived fuel?
Reply: The estimation of the costs of a pyrolysis process is not an easy task since in Europe there are few commercial-scale bio-oil production plants (Table 1 Giudicianni et al. 2021). The only pyrolysis plant operating on an industrial scale (TRL 9) is located in Netherlands, it is managed by the Twence Hengelo company and it use biomasses (about 24kton / year) as feedstock. There are also two demonstration plants (TRL 6-7), also for pyrolysis of biomass, in Finland and Germany managed by Fortum Joensuu and the 50 and 24 kton / year KIT respectively. Two 24kton / year industrial plants are under construction for biomass in Finland and Sweden, while two plants are being designed, in Norway and Germany, of which the largest is 100kton / year (the Norwegian one).
S.-K. Ning et al. (Journal of Cleaner Production 59 (2013) 141e14) report a specific cost of pyrolysis oil production of NT$11,758/m3 (NT$1 = US$0.034 in October 2012) but also in this case biomass is the feedstock used for the production of oil. As concerns the gasification process, U. Arena et al. (Fuel Processing Technology 131 (2015) 69–77) report a cost of the capital expenditure (CapEx) of 485 €/(t/y) and a total cost of operation (OpEx) of 373 k€/y for an air gasification plant (a 400 kWe plant) for energy recovery from solid recovered fuel (SRF).
Umberto Arena, Fabrizio Di Gregorio, Gianfranco De Troia, Alessandro Saponaro A techno-economic evaluation of a small-scale fluidized bed gasifier for solid recovered fuel Fuel Processing Technology 131 (2015) 69–77
S.-K. Ning et al. Journal of Cleaner Production 59 (2013) 141e14)
Point 2: suggest to add some performance comparison with other similar studies in the field, especially regaridng the real engineering application
Reply: As reported in the answer to point 1, it is not easy to make a comparison with real plant performances because no data on industrial scale is available mostly because the pyrolysis process is currently not fully developed on industrial scale and generally the existing plants process biomass-based feedstocks and not refused fuels.
As concerns the gasification test, we can say that our data is very close with that reported by Arena et al. 2015 (normalized on N2 free basis) obtained processing a real RDF in a pilot scale bubbling fluidized bed reactor under conditions comparable with those adopted in this paper. It is worth of nothing that a lower concentration of tar is obtained in our experiment. For sake of completeness, below we report a table in which the composition of syngas reported in Arena et al 2015 is compared with that of the syngas obtained during our experiment (both syngas compositions are normalized on N2 free basis)
|
Arena et al. 2015 |
Arena et al. 2015 syngas free of N2 |
This paper |
|
|
co2 |
12,83 |
33,64 |
27,18 |
|
co |
10,4 |
27,27 |
31,31 |
|
h2 |
8,24 |
21,60 |
15,96 |
|
ch4 |
4,35 |
11,41 |
12,97 |
|
c2h4 |
2,05 |
5,37 |
4,01 |
|
c2h6 |
0,04 |
0,10 |
0,1 |
|
c2h2 |
0,16 |
0,42 |
0,01 |
|
c3h6 |
0,02 |
0,05 |
7,83 |
|
c6h6 |
0,1 |
0,26 |
The paper Arena et al 2015 has been added as reference in the revised manuscript.
Reviewer 3 Report
I congratulate the authors, the article is good and only a little bit left until publication! Just some small comments from my side.
It is generally accepted that the Materials and Methods section precedes the Results and Discussion. I invite you to specify and change the title 3.4. Methods.
The methods used should be explained in more detail so that those who are not experts in the field can understand.
After the conclusions, it is necessary to put in some concrete proposals. As well, recommendations for further research are welcome.
supporting information can't be downloaded at 714 www.mdpi.com/xxx/s1,File not found
Author Response
Reviewer#3
Comments and Suggestions for Authors
I congratulate the authors, the article is good and only a little bit left until publication! Just some small comments from my side.
We thank the Reviewer for his/her positive evaluation of our work.
Point 1: It is generally accepted that the Materials and Methods section precedes the Results and Discussion. I invite you to specify and change the title 3.4. Methods.
Reply: We agree with the Reviewer preference, but, unfortunately, the template provided by the Journal requires that the experimental section comes after the Results and Discussion section. In accordance with the Reviewer’ suggestion, the title of section 3.4 has been changed in Analytical techniques
Point 2: The methods used should be explained in more detail so that those who are not experts in the field can understand.
Reply: We thank the Reviewer for the advice, the section 3.4 has been revised accordingly and additional details about the analytic techniques applied and the outcome of the measurements have been added.
Point 3: After the conclusions, it is necessary to put in some concrete proposals. As well, recommendations for further research are welcome.
Reply: We thank the Reviewer for the kind suggestion, Conclusions section has been revised accordingly.
Point 4: supporting information can't be downloaded at 714 www.mdpi.com/xxx/s1,File not found
Reply: We apology for the problem occurred.
Round 2
Reviewer 1 Report
AGREE